# The Mechanisms of Restenosis and Relevance to Next Generation Stent Design

**DOI:** 10.3390/biom12030430

**Published:** 2022-03-10

**Authors:** Jessie Clare, Justin Ganly, Christina A. Bursill, Huseyin Sumer, Peter Kingshott, Judy B. de Haan

**Affiliations:** 1Department of Chemistry and Biotechnology, Swinburne University of Technology, Melbourne, VIC 3122, Australia; jclare@swin.edu.au (J.C.); jganly@swin.edu.au (J.G.); pkingshott@swin.edu.au (P.K.); 2Baker Heart and Diabetes Institute, Melbourne, VIC 3004, Australia; 3Adelaide Medical School, Faculty of Health and Medical Sciences, University of Adelaide, Adelaide, SA 5000, Australia; christina.bursill@sahmri.com; 4Vascular Research Centre, Lifelong Health Theme, South Australian Health and Medical Research Institute, Adelaide, SA 5000, Australia; 5ARC Centre of Excellence for Nanoscale BioPhotonics, Adelaide, SA 5000, Australia; 6ARC Training Centre in Surface Engineering for Advanced Materials (SEAM), Swinburne University of Technology, Melbourne, VIC 3122, Australia; 7Department Cardiometabolic Health, University of Melbourne, Melbourne, VIC 3010, Australia; 8Department of Physiology, Anatomy and Microbiology, La Trobe University, Melbourne, VIC 3086, Australia; 9Department of Immunology and Pathology, Central Clinical School, Monash University, Melbourne, VIC 3004, Australia

**Keywords:** restenosis, neointimal hyperplasia, drug-eluting stents, inflammation, redox

## Abstract

Stents are lifesaving mechanical devices that re-establish essential blood flow to the coronary circulation after significant vessel occlusion due to coronary vessel disease or thrombolytic blockade. Improvements in stent surface engineering over the last 20 years have seen significant reductions in complications arising due to restenosis and thrombosis. However, under certain conditions such as diabetes mellitus (DM), the incidence of stent-mediated complications remains 2–4-fold higher than seen in non-diabetic patients. The stents with the largest market share are designed to target the mechanisms behind neointimal hyperplasia (NIH) through anti-proliferative drugs that prevent the formation of a neointima by halting the cell cycle of vascular smooth muscle cells (VSMCs). Thrombosis is treated through dual anti-platelet therapy (DAPT), which is the continual use of aspirin and a P2Y_12_ inhibitor for 6–12 months. While the most common stents currently in use are reasonably effective at treating these complications, there is still significant room for improvement. Recently, inflammation and redox stress have been identified as major contributing factors that increase the risk of stent-related complications following percutaneous coronary intervention (PCI). The aim of this review is to examine the mechanisms behind inflammation and redox stress through the lens of PCI and its complications and to establish whether tailored targeting of these key mechanistic pathways offers improved outcomes for patients, particularly those where stent placement remains vulnerable to complications. In summary, our review highlights the most recent and promising research being undertaken in understanding the mechanisms of redox biology and inflammation in the context of stent design. We emphasize the benefits of a targeted mechanistic approach to decrease all-cause mortality, even in patients with diabetes.

## 1. Introduction

An estimated 18 million lives are lost each year as a consequence of cardiovascular diseases (CVD), making it the primary cause of death worldwide [1]. One of the major CVDs is coronary artery disease (CAD). CAD is characterised by the deposition of lipids, such as cholesterol, into the arterial wall. This process is known as atherosclerosis and can lead to partial or total occlusion of the vessel, causing cardiac ischemia and potentially leading to a myocardial infarction [2,3,4]. These potentially life-threatening events require urgent medical treatment. Early restoration of blood flow to the heart muscle via reperfusion of the coronary arteries is key to the prevention of cardiac cell death and left ventricular remodelling. One of the main non-surgical treatments currently available is percutaneous coronary intervention (PCI) [5], where a mechanical device or stent is inserted into the occluded vessel with the aid of an inflatable catheterisation balloon (balloon angioplasty) to force it to remain open.

Throughout the last two decades, numerous advances have been made in PCI with the advent of drug-eluting stents (DES), which release anti-proliferative drugs such as sirolimus. Contemporary DES have provided better outcomes for patients when compared with the older bare-metal stents (BMS); however, stent failure due to restenosis or thrombosis still poses a challenge, with the incidence of in-stent restenosis (ISR) estimated to be between 5 and 10% [6,7]. Restenosis is characterised by an occlusion in the lumen of the vessel and is considered significant when a ≥50% reduction in the luminal vessel diameter occurs after PCI [6]. Blood clot formation within the stented region is known as stent thrombosis (ST), which also poses a substantial risk when using a DES, for which newer drug treatments are currently being investigated [8]. Despite the current reduced restenosis rates, patients with comorbidities such as diabetes mellitus (DM) remain at a significantly elevated risk of ISR or ST, potentially enhanced by the pathological state within the arteries [9,10]. Indeed, Wang et al. report an ISR rate of approximately 20% (74 out of 368 patients), suggesting a 2–4-fold greater risk for diabetic patients [9]. Given that patients living with type 2 diabetes mellitus (T2DM) are expected to increase from 463 million, as assessed in 2019, to 700 million patients worldwide by 2045 [11], there remains a significantly large number of patients at risk for ISR and ST. Thus, investigating improved stent designs, including the incorporation of new drugs, is crucial in aiding the treatment of CAD in patients with comorbidities such as DM, where elevations in blood glucose levels (hyperglycaemia) are known to drive redox and inflammatory pathways [12].

In designing more effective stents, it is important to consider the underlying pathology that drives restenosis and thrombosis with a clear focus on the mechanisms involved. This will facilitate the development of novel stent coatings and biomaterials that target these key drivers of ISR and ST. Evidence points to a major role of inflammatory mediators and redox-driven processes as significant contributors to ISR and ST. Understanding the processes involved and designing mechanism-based coatings and biomaterials is more likely to yield efficacious outcomes for patients at risk of stent failure who require life-saving PCI. Indeed, preclinical studies are now focusing on coatings containing agents such as endothelial-like bioactive molecules, antioxidants or antibodies that could potentially reduce restenosis or thrombosis through targeting specific pathways [13]. Many of these preclinical investigations have yielded positive results; however, further investigation is needed to determine if some of these more recent stent designs can be translated to the clinic.

This review will examine the involvement of inflammation and redox processes as key drivers of ISR and ST. It will also discuss the most commonly used stents according to market share and compare their pros and cons regarding restenosis and thrombosis prevention, as well as the future directions of stent research. In doing so, we highlight the progress that has been made to date, yet offer new alternative approaches that circumvent some of the key issues that limit the use of current DES.

## 2. Mechanisms of Restenosis

Restenosis is a complex phenomenon that is generally defined as a reduction in vessel lumen diameter after PCI, though it can be more clearly defined in two ways. Clinically, restenosis is defined as the recurrence of angina symptoms which may then require revascularisation via PCI or coronary artery bypass grafting (CABG) if the stenosis is too advanced [14]. On the other hand, angiographic restenosis is described as the re-narrowing of the lumen to ≥50% occlusion after PCI, typically within a follow-up period of 3–6 months [15,16]. The underlying mechanisms of the disease are yet to be fully elucidated, but restenosis is believed to develop via the following steps: denudation of the endothelial cells (ECs) during balloon angioplasty and stenting, initiation of inflammatory and redox processes, followed by neointimal hyperplasia and/or neoatherosclerosis [15].

### 2.1. Balloon-Mediated Endothelial Denudation

An intact and healthy vascular endothelium will produce and release two key vasoactive molecules, nitric oxide (NO) and prostacyclin (prostaglandin I_2_; PGI_2_). NO is a gaseous free radical that was first described as a mediator of vasodilation in 1987 [17,18,19]. It also has antioxidant, anti-platelet and anti-inflammatory properties, which are atheroprotective in nature [20]. PGI_2_ is a potent vasodilator that was first reported in 1976 [21]. Apart from this property, PGI_2_ is also an inhibitor of platelet aggregation and works synergistically with NO to modulate platelet activation and aggregation during vascular inflammation [21,22].

During stent placement, the arterial wall is mechanically injured and denuded of its tunica intima, which is comprised of endothelial cells. This exposes the vascular smooth muscle cells (VSMCs) and extracellular matrix (ECM) of the tunica media to the stent surface and the circulating blood components. With fewer endothelial cells present, less NO and PGI_2_ is produced and secreted, causing a decrease in the anti-platelet activity of the remaining competent ECs [23]. In combination, these steps lead to the rapid deposition of platelets and fibrinogen (beginning of thrombosis) at the site of injury, which marks the beginning of vascular inflammation (see Figure 1) [24]. For patients with DM, the denudation of competent ECs can have a worsened effect due to the reduced bioavailability of NO in the presence of hyperglycaemia [12,25]. This state of endothelial dysfunction promotes the attachment of platelets and leukocytes to the stented region, contributing to the progression of vascular inflammation [12].

### 2.2. Vascular Inflammation

Vascular inflammation involves complex interactions between numerous cell types that release pro-inflammatory markers, cytokines, chemokines, and/or express cellular adhesion molecules (CAMs). Immediately following PCI, the surrounding endothelial cells are “activated” by pro-inflammatory cytokines such as interleukin-1 beta (IL-1β), interleukin-6 (IL-6) and tumour necrosis factor-alpha (TNFα) secreted by monocytes and underlying VSMCs [26] (refer to Figure 1). Subsequently, the ECs upregulate the expression of CAMs, including intercellular adhesion molecule-1 (ICAM-1), vascular cell adhesion molecule-1 (VCAM-1) and E-selectin, with P-selectin being translocated from Weibel–Palade bodies to the endothelial cell membrane [22,26]. This sets in motion an ongoing inflammatory response whereby circulating monocytes are recruited to the site of injury. In vivo studies show that monocytes begin rolling and attach firmly to ECs via the interaction of β2 integrins, Mac-1 (CD11b/CD18) and lymphocyte function-associated antigen-1 (LFA-1) with ICAM-1 [27,28]. In regard to post-PCI, higher levels of Mac-1 expression in neutrophils have been associated with late lumen loss and an increased risk of restenosis through neointimal hyperplasia [24,28]. The interaction of Mac-1 and LFA-1 with ICAM-1 facilitates the transendothelial migration of monocytes into surrounding tissues, a key step in the process of inflammation, one of the drivers of restenosis [24,27,28]. This will be discussed more in depth in Section 2.4.

Nuclear factor-kappa beta (NF-κB) is a family of proteins whose role as a transcription factor is to modulate the expression of numerous target genes, many of which are involved in inflammatory processes; thus, NF-κB is said to be a mediator of inflammation [29]. Briefly, the canonical NF-κB activation pathway is triggered by cytokines IL-1 and TNFα, which causes a multitude of intracellular interactions that lead to the translocation of NF-κB to the nucleus and binding with DNA (see reviews [29,30] for more detail). Apart from cytokine activation, NF-κB can also be activated by toll-like receptors (TLRs).

Toll-like receptors are transmembrane proteins, known as pattern recognition receptors, that play a vital role in pathogen identification and host defence [31]. An important component of TLR proteins is the cytoplasmic Toll/IL-1 receptor (TIR) domain, which is similar to that of the IL-1 receptor and thus enables similar signalling targets such as NF-κB [32]. TLR-mediated activation of NF-κB occurs via adapter proteins that lead to the phosphorylation of inhibitor proteins called IκBs, which are responsible for preventing the translocation of NF-κB to the nucleus [33]. Once phosphorylated, IκB proteins are ubiquitinated and degraded via proteasomes, thus allowing for the translocation of NF-κB to the nucleus and subsequent binding to DNA as mentioned above [32,33]. Interestingly, many target genes of NF-κB induce chemokine and cytokine expression, thereby participating in a positive feedback loop that contributes to vascular inflammation and promotes restenosis [26]. Through paracrine signalling, activated ECs can upregulate the activity of NF-κB in surrounding cells such as VSMCs, potentially contributing to phenotype switching, proliferation, and migration of VSMCs into the subintimal space, triggering the onset of neointimal expansion [26,29]. In a stented hypercholesterolaemic rabbit model, a decoy oligodeoxynucleotide inhibited NF-κB binding to the promoter region of pro-inflammatory genes such as MCP-1 (monocyte chemoattractant protein-1), IL-1β and IL-6, which indirectly led to a reduction in the neointima of the stented vessel [34]. This study suggests a possible mechanism whereby NF-κB action in ECs may drive inflammatory mediated changes to VSMCs.

Patients with hyperglycaemia, such as those with T2DM, have chronic inflammation as evidenced by elevated serum levels of inflammatory cytokines IL-6 and TNFα [35]. In vitro studies in human umbilical vein endothelial cells (HUVECs) have demonstrated that under hyperglycaemic conditions, the expression of cellular adhesion molecules (VCAM-1, ICAM-1 and E-selectin) [36] and a chemokine (MCP-1/CCL2) were upregulated [37]. These findings were also confirmed with in vivo studies [38]. Furthermore, these chemoattractant and adhesive molecules aid in driving transendothelial migration of monocytes into the subintima, as well as the induction of inflammatory stimuli. In particular, IL-6 and its signalling pathways have been shown to mediate vascular complications in T2DM as well as VSMC proliferation [39]. Thus, there may be a link between DM and the excessive proliferation of VSMCs, which in turn play a role in the development of neointimal hyperplasia.

Chemokines are small proteins involved in the inflammatory response. Some such as MCP-1 (CCL2) [40], RANTES (CCL5) [41] and fractalkine (CX_3_CL1) [42], have been implicated in the development of neointimal hyperplasia through VSMC proliferation and migration (see review [43] for further detail). Particularly, antibody blockade of MCP-1 and its receptor CCR2 attenuated neointimal hyperplasia in balloon-injury preclinical models, thus highlighting the important role that MCP-1/CCR2 plays in inflammation and neointimal tissue formation [43,44]. Another study using a cuff-induced arterial injury model demonstrated that 21 days after injury, there was a significant decrease in intimal area, intima/media ratio and percent luminal stenosis in mice deficient in CCR2 when compared to CCR2^+/+^ mice [40]. Taken together, these studies, along with the knowledge that MCP-1 is required for monocyte chemotaxis, suggest that MCP-1/CCR2 plays a key part in neointimal hyperplasia (NIH). To potentially limit inflammation and restenosis after stenting, future research could be centred on developing drugs that block the MCP-1/CCR2 axis, in turn reducing neointimal proliferation [43].

It is also highly likely that perivascular adipose tissue (PVAT), which is known to surround most blood vessels [45], plays a role in driving inflammation after PCI. PVAT is a connective tissue comprised of a mix of cell types, including adipocytes, preadipocytes, mesenchymal stem cells, fibroblasts, vascular and nerve cells, and, importantly, inflammatory cells (macrophages, lymphocytes and eosinophils) [46]. Under pathological conditions, PVAT expands, becomes dysfunctional and initiates inflammatory crosstalk with the blood vessels [46]. TNF-α, IL-6, IL-1β and MCP-1 levels are increased in PVAT associated with atherosclerosis [47,48]. In particular, PVAT of the coronary arteries (C-PVAT) shows a more inflamed phenotype compared with non-coronary PVAT [49]. The involvement of an inflamed PVAT is exacerbated by risk factors such as obesity [50], a major underlying pathology driving T2DM. Indeed, PVAT has been shown to be highly inflammatory in T2DM patients [51] and is likely to be the mechanistic link between T2DM and atherosclerosis [52]. Inflammation also drives diabetes-mediated endothelial dysfunction, a further risk factor for developing atherosclerosis. Thus, the linking together of a triad of pathologies (obesity, diabetes and endothelial dysfunction) by elevated PVAT inflammatory processes points to the value in evaluating the role of PVAT on processes such as PCI-driven neoatherosclerosis.

### 2.3. Redox Processes, Endothelial Dysfunction and the Development of Restenosis

Oxidative stress is generally defined as an imbalance between pro-oxidant reactive oxygen species (ROS) and their antioxidant counterparts. Excessive production of ROS such as superoxide (O_2_^−^), peroxynitrite (ONOO^−^) and hydrogen peroxide (H_2_O_2_) is associated with the pathogenesis of several cardiovascular diseases, such as atherosclerosis [53]. Additionally, evidence points towards oxidative stress playing a vital role in the development and progression of diabetic vasculopathies [53]. In the vasculature, superoxide is the main ROS produced. There are several key pathways that can generate O_2_^−^–NADPH oxidases (Nox), mitochondrial electron transport chain and eNOS uncoupling [54].

NADPH oxidases (Nox) have several different isoforms in the vasculature, including Nox1, Nox2, Nox4 and Nox5. Nox catalyses the generation of O_2_^−^ via the reduction of molecular oxygen using NADPH or NADH as the electron donor [54] (refer to Figure 2). Nox1 expression is induced by growth factors, cytokines and other factors that may be elevated under pathological conditions. In a balloon-injury model of the vasculature, Nox1-deficient mice were shown to have reduced VSMC proliferation and migration compared to WT and Nox1-overexpressing mice [55]. In another study, there was an increase in the expression of Nox1, oxidative stress and pro-inflammatory markers in human aortic endothelial cells (HAECs) treated with high glucose [56]. Additionally, this study demonstrated that Nox1 deletion was associated with a reduction in ROS production as well as lessening the expression of pro-inflammatory markers and CAMs under diabetic conditions [56]. When taken together, these studies demonstrate that Nox1 is vital in the development of neointimal tissue formation, including under hyperglycaemic conditions, and is essential for the production of ROS in the vasculature. At present, the targeting of Nox enzymes is limited to dual inhibitors such as GKT137831, which target both Nox1 and Nox4. This molecule, through inhibition of Nox1/Nox4, was shown to decrease atherosclerosis under diabetic conditions [56]. Other examples of Nox1-inhibiting molecules that may have potential therapeutic use are ML171, a Nox1 inhibitor [57], and NoxA1ds, an inhibitor of the Nox1 activator subunit domain NoxA1 [58]. Nevertheless, there is a lack of clear evidence as to whether these molecules could be efficacious in vivo for attenuating vascular complications after stenting, though this could provide an interesting direction for future work.

In endothelial cells, NO plays a key role in maintaining vascular tone and anti-inflammatory and antioxidant effects along with the inhibition of VSMC migration and proliferation. Under physiological conditions, endothelial nitric oxide synthase (eNOS) catalyses the conversion of l-arginine into l-citrulline and NO in the presence of (*6R-*)5,6,7,8-tetrahydro-l-biopterin (BH_4_) and NADPH as co-factors [59]. NO is then secreted from the ECs and binds to soluble guanylate cyclase (sGC) in VSMCs to contribute to the modulation of vascular tone [60]. Briefly, this occurs through the conversion of GTP to the secondary messenger cGMP, which then activates cGMP-dependent protein kinase G (PKG) to phosphorylate its target receptor, increasing Ca^2+^ efflux, which leads to VSMC relaxation [60].

Endothelial dysfunction is a term used to describe the inability of ECs to maintain vascular homeostasis. The exact causative mechanisms of this dysfunction are unclear, although it is suggested that increased ROS production and eNOS uncoupling are key factors (as shown in Figure 2). Superoxide can scavenge NO to form peroxynitrite (ONOO^−^), thereby reducing the bioavailability of NO and leading to eNOS uncoupling [54]. Instead of NO, eNOS produces O_2_^−^ which can lead to further ONOO^−^ formation or, via catalysis with the enzyme superoxide dismutase (SOD), will be converted to hydrogen peroxide (H_2_O_2_) [54]. Through the enhanced production of ROS, NO is inactivated, and its protective effects, such as reducing platelet/leukocyte adhesion and VSMC growth and migration, are limited [61].

**Figure 2 biomolecules-12-00430-f002:**
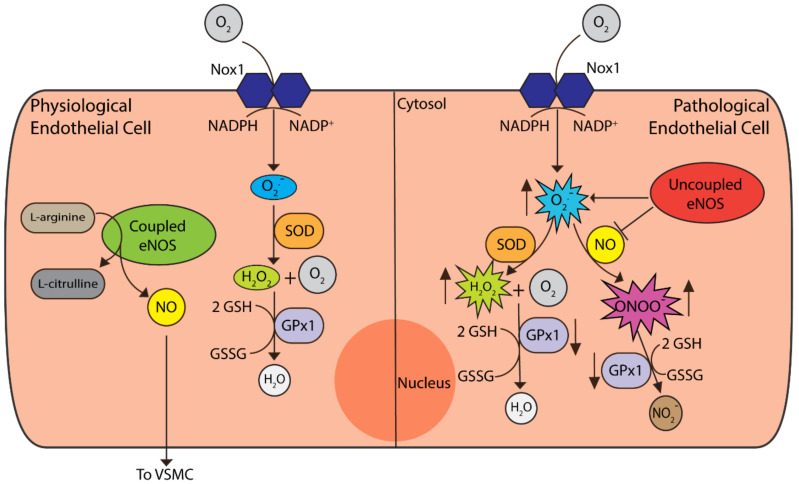
Redox processes in endothelial cells (ECs). Under physiological conditions, coupled eNOS synthesises nitric oxide (NO) and l-citrulline from l-arginine, with NO secretion and interaction with VSMCs. Nox1 is the predominant isoform of the Nox enzymes responsible for superoxide (O^−^) production, required for physiological processes. Superoxide levels are controlled by the superoxide dismutases (SOD) that convert superoxide to hydrogen peroxide (H_2_O_2_), which is then neutralised to water by glutathione peroxidase (GPx) enzymes. GPx1 is the most abundant cytosolic isoform shown to play an important role in ECs [62]. Under pathological conditions, uncoupled eNOS no longer produces NO and instead synthesises superoxide. Superoxide then interacts with NO to form peroxynitrite (ONOO^−^), thus reducing the bioavailability of NO. Apart from hydrogen peroxide neutralisation, GPx1 can also convert peroxynitrite to nitrite (NO_2_^−^). Under pathological conditions, such as in the setting of diabetes, the activity of GPx1 declines. This, in turn, leads to an increase in reactive oxygen species (ROS) such as H_2_O_2_, ONOO^−^ and lipid peroxides which promote endothelial dysfunction.

Apart from NO production by eNOS, NO can also be stored in, and released from, small molecules named *S*-nitrosothiols (RSNOs). These molecules are called NO donors due to their ability to enzymatically break down and release NO into the surrounding tissue. These NO donors can be synthesised endogenously, with some, such as *S*-nitrosoglutathione (GSNO), reported to enable NO storage in the vascular wall [63,64]. Similarly, synthetic RSNOs such as *S*-nitroso-*N*-acetyl-penicillamine (SNAP) have also been demonstrated to provide a depot of NO in the vascular wall along with the ability to catalytically generate NO [63,64,65] (refer to Table 1). Due to these properties, NO donors such as SNAP and *S*-nitroso-*N*-acetylcysteine (NACNO) are currently undergoing investigation to determine if they could be used therapeutically to treat endothelial dysfunction [63] and may additionally function to limit restenosis post PCI.

Antioxidant functioning molecules such as vitamins E and C, β-carotene, *N*-acetylcysteine (NAC), probucol and resveratrol have all been investigated as potential treatments for oxidative stress and cardiovascular disease in general. Meta-analyses of large-scale randomised clinical trials showed that vitamin E, vitamin C and β-carotene were not able to confer a therapeutic effect to CVD morbidity or mortality [66,67]. The conclusions stated that vitamin E, vitamin C and β-carotene are not currently recommended as treatments. Another smaller study showed that probucol—but not vitamins E, C or β-carotene—had some positive effect in reducing restenosis after PCI [68]. However, one limitation to this study and the treatment potential of this drug is the administration time frame. The beneficial effect was seen after 30 days of treatment pre-PCI and then 6 months post-PCI. This may limit the therapeutic use of this antioxidant as 30 days pre-treatment is not possible in acute or urgent cases of PCI. A similar finding was found in a preclinical study using NAC, whereby neointimal hyperplasia was not significantly reduced between the control and treatment groups [69]. More recently, however, the polyphenol resveratrol has shown promising results in reducing neointimal hyperplasia either on its own or in combination with other therapies [70,71,72]. Despite the lack of clear evidence if nutritional antioxidants may provide protective effects against restenosis, some such as resveratrol provide a promising avenue for further research that could be used in conjunction with more defined therapies. In contrast, endogenous antioxidant enzymes may be a better target in ameliorating oxidative stress in the vasculature.

Endogenous antioxidant enzymes are important in the regulation of ROS production and play a protective role in the vasculature. Two of the key antioxidant enzyme families used to mitigate ROS damage are the superoxide dismutases (SOD) and the glutathione peroxidases (GPx). O_2_^−^ is converted into water in a two-step reaction. First, SOD catalyses the dismutation of O_2_^−^ into H_2_O_2_ and molecular O_2_, and then GPx-family members, via reduced glutathione (GSH), reduce H_2_O_2_ into water [54]. Apart from this reaction, GPx also reduces other ROS such as ONOO^−^ to nitrite (NO_2_^−^), and lipid peroxide (LOOH) is reduced to lipid alcohol (LOH) and water [73].

The GPxs are a family of selenoenzymes that catalyse the breakdown of H_2_O_2,_ LOOH and ONOO^−^ via the reduction of glutathione (GSH) to oxidised glutathione (GSSG). The most abundant and cytosolic isoform is glutathione peroxidase-1 (GPx1). Studies in mice have shown that a lack of GPx1 leads to a pro-inflammatory vascular milieu that can potentiate other dysregulatory mechanisms [38,62,74]. Decreased GPx1 activity has also been shown in excised human carotid arteries [75]. More recently, therapeutic mimetics of GPx1 such as the organoselenium compounds selenocystamine (SeCA), 3,3′-diselenodipropionic acid (SeDPA) and ebselen (2-phenyl-1,2-benzoselenazol-3(2*H*)-one) have been reported to reduce oxidative stress through the catalytic generation of NO from RSNOs [64,65,76]. These studies, amongst others, demonstrate the potential therapeutic benefit of GPx1 mimetics in mitigating oxidative stress pathways. Apart from its role in oxidative stress, GPx1 is also involved in regulating reductive stress.

Reductive stress is the counterpart to oxidative stress and is characterised by an increase in reducing equivalents (NADH, NADPH and GSH) that form the redox couples NADH/NAD^+^, NADPH/NADP^+^ and GSH/GSSG [77]. In particular, reduced glutathione (GSH) is significantly increased, contributing to cytotoxicity via *S*-glutathionylation of proteins. Indeed, patients with T2DM who have microvascular disease display increased levels of *S*-glutathionylated haemoglobin compared to controls [78]. Interestingly, in some instances, reductive stress, in the form of high NADH to NAD^+^ ratios in mitochondria, is also known to promote ROS formation to a level that exceeds the ROS scavenging capability of antioxidants [79]. At physiological levels, ROS are important signalling molecules, and reductive stress can attenuate these pathways by a reduction in ROS, which can then lead to further downstream effects [80]. Despite the increasing evidence that reductive stress is as important, if not more so than oxidative stress, there has been less focus placed on reductive stress, and as outlined below, it could represent a key area for further investigation.

Using a balloon-injury atherosclerosis-prone animal model, Ali et al. [81] demonstrated that a loss of GPx1 alters redox stress in VSMCs and promotes their proliferation and migration both in vitro and in vivo [81]. They discovered that ROS1, a protooncogene receptor tyrosine kinase, mediated these effects in GPx1-deficient VSMCs. Interestingly, they showed that ROS1 was not directly affected by reductive stress, but instead, it inhibited a phosphatase, SHP-2, by *S*-glutathionylation [81]. This modification of SHP-2 was then shown to prevent dephosphorylation of ROS1, which led to the initiation of VSMC proliferation and neointimal hyperplasia. Overall, this study highlights the need for further research to investigate reductive stress as a mediator of neointimal hyperplasia and vascular remodelling.

### 2.4. Neointimal Hyperplasia (NIH)

Stenting results in EC denudation that triggers inflammation resulting in signalling molecules being released into the surrounding tissue with a multitude of effects. This causes the four principal interlinked factors which all lead to neointimal hyperplasia: the transformation of vascular smooth muscle cells, monocyte build-up in the subintimal space, endothelial damage, and migration of fibroblasts. One effect is the phenotypic switching of VSMCs from a quiescent contractile state to a proliferative ‘synthetic’ state. These modulated VSMCs gain phenotype-specific functionalities such as proliferation, migration, and further extracellular matrix synthesis [28,82]. This switching is a result of multiple factors such as cyclic stretching stimulating the MEF2B pathway. This leads to NADPH oxidase isoform 1 (Nox1)-derived ROS generation, which contributes to phenotype switching by altering the cytoskeletal structure of the VSMC, namely the reduction of contractile proteins calponin 1 (CNN1) and smoothelin B [83]. Other factors include those present during an inflammatory response, growth factors such as transforming growth factor β1 (TGF-β1) and platelet-derived growth factor (PDGF). TGF-β1 has been shown to activate the P13K/AKT/ID2 pathways, which cause cell migration, proliferation and the production of extracellular matrix indicative of NIH [84]. TGF-β acts on SMCs by activation of Smad proteins, causing phenotype switching leading to NIH [85]. This is typically demonstrated by the activation of genes in mature SMCs such as αSMA, SM22α and SMMHC, causing differentiation [86]. While TGF-β has been shown to induce proliferation at lower concentrations (0.025 ng/mL), at higher concentrations (0.1 ng/mL), it has been observed to attenuate VSMC proliferation [87]. This interaction is also complicated by the presence of other growth factors; for instance, treatment of VSMCs with TGF-β in 1% FBS had no effect on cell number, while 5% induced proliferation [88]. Not only does TGF-β have an effect on SMCs but also on ECs, causing them to become more mesenchymal-like in nature, also known as the endothelial to mesenchymal transition (Endo-MT) [89]. The endo-MT process is also modulated by shear stress commonly present following stent implantation [90]. This contributes to NIH as the endothelial characteristics of the cells are lost, such as anti-thrombogenicity [91], and the endothelial cells also act as a source for smooth muscle-like cells to form the neointima and add to the fibrosis process of neointimal formation [92]. PDGF has been shown to induce VSMC migration by binding to PDGF β receptors [93]. This triggers the production of extracellular matrix by induction of lysyl oxidase (LO), thrombospondin, osteopontin and the intracellular enzyme lactate dehydrogenase (LDH) [94]. This induces the expression of MCP-1 and other monocyte chemoattractants [95]. Despite the attraction of monocytes to the site of injury and their role in NIH, VSMCs are still considered the main contributor to neointimal tissue formation.

Monocyte build-up in the subintimal space is a process whereby monocytes traverse the endothelium and occurs in a stepwise fashion. The first steps are rolling and capturing monocytes that are attracted to endothelial cell surfaces by chemokines. Monocytes are captured by L-, P- and E- selectin expressed by activated endothelial cells, which then bind to P-selectin glycoprotein 1 (PSGL-1) and peripheral lymph node addressin (PNAd) [96]. Activated endothelial cells also express ICAM-1 and VCAM-1 [97], which additionally facilitate monocyte rolling and adhesion. Transmigration of monocytes into the subintima is then facilitated by junction-associated molecules (JAM), which are ligands of β_2_ integrin LFA-1 [98]. Within the subintimal space, monocytes differentiate into macrophages, releasing cytokines such as IL-1β and exerting a further inflammatory effect on the endothelial cells, thereby increasing their permeability. This leads to low-density lipoproteins (LDL) entering the subintimal space, where they are taken up by macrophages to become foam cells in a similar fashion to the atherosclerotic process [99]. While endothelial dysfunction and damage is the primary driver behind NIH, adventitial fibroblasts are also recognised to play a significant role in NIH [100].

Fibroblast migration is the final major contributing factor to NIH. VSMCs involved in the NIH cascade produce growth factors such as TGF-β1, enhancing the migration and proliferation of not only VSMCs but fibroblasts present in the adventitia [101]. This causes the fibroblasts to undergo a transformation into myofibroblasts [102]. These cells then contribute to the formation of ECM by increasing protein synthesis of molecules such as collagen and fibronectin. Since these cells also acquire the characteristics of smooth muscle cells via the production of α-smooth muscle actin (αSMA), they can also migrate into the media, further decreasing the vessel lumen diameter [103].

As a consequence of these newly acquired functionalities, the intimal layer of the vascular wall increases and consists mainly of VSMCs, ECM components, some fibroblasts and foam cells. This hyperplasia is strongly correlated with inflammation triggered by vascular injury and causes a thickening that reduces the patency of the stent and overall lumen diameter, ultimately leading to the development of restenosis [104]. While a major complication arising from stenting, NIH has been shown to be treatable by anti-proliferative drugs, which stall VSMC proliferation and migration into the subintimal space. This treatment is discussed further in Section 4.1.2 and Section 4.1.3, and two of the key limus-based drugs are described in Table 1.

**Table 1 biomolecules-12-00430-t001:** Chemical structure and target of clinical and preclinical therapeutic drugs in stenting.

Drug Name	Structure	Mechanism of Action	Reference
**-limus based**			
Sirolimus	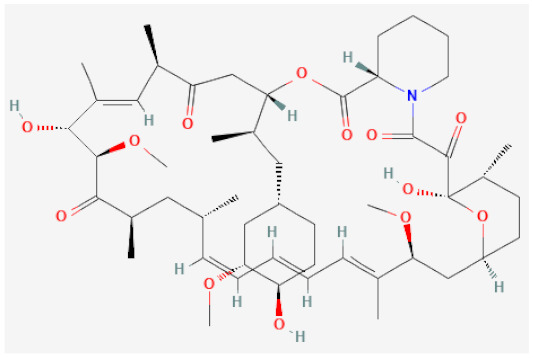	Inhibitor of mechanistic target of rapamycin (mTOR), cell cycle arrest	[105,106]
Everolimus	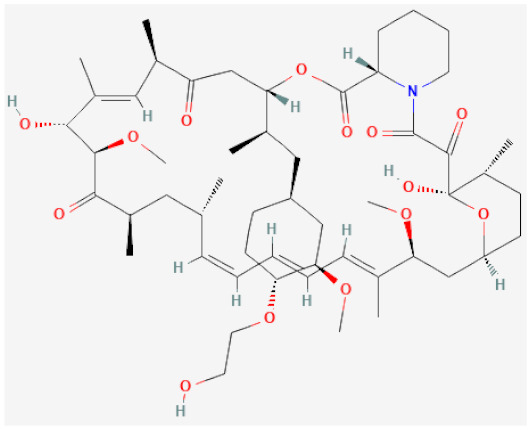	Inhibitor of mechanistic target of rapamycin (mTOR), cell cycle arrest	[107,108]
**Organoselenium based**			
Selenocystamine	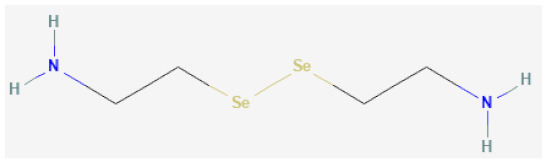	GPx1 mimetic, acts via selenium to catalytically generate NO from RSNOs	[65,109,110]
Ebselen	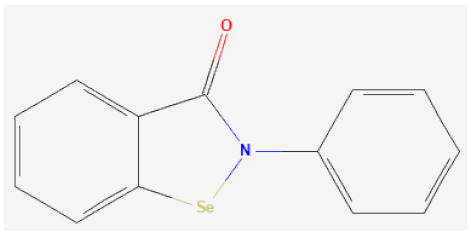	GPx1 mimetic, acts via selenium to lessen redox stress	[111,112,113]
**NO donors**			
*S*-nitroso-*N*-acetylcysteine (NACNO)	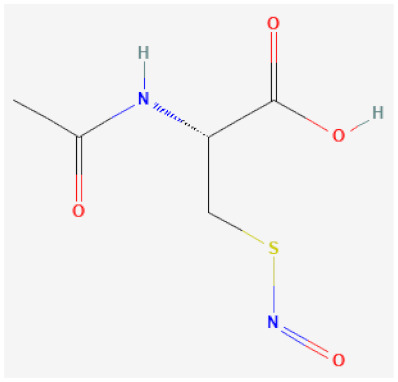	Catalytically decomposes through Se species interaction, releases NO	[63,114]
*S*-nitroso-*N*-acetylpenicillamine (SNAP)	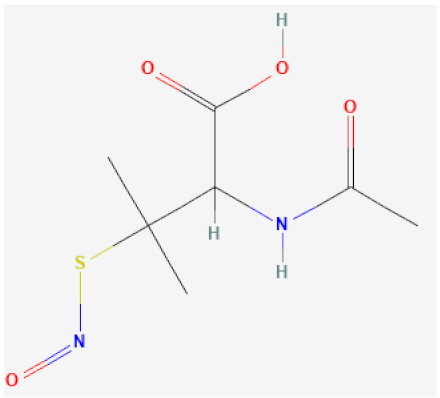	Catalytically decomposes through Se species interaction, releases NO	[63,115,116]

### 2.5. Neoatherosclerosis

While neointimal hyperplasia has largely been mitigated using anti-proliferative drugs, neoatherosclerosis, which occurs as a late complication (2–7 years post stenting) of PCI, remains a significant complication and a potential threat to patient health. Neoatherosclerosis is the formation of a new plaque inside the stent, with mechanisms believed to occur in a similar manner to native atherosclerosis, but on a shorter time scale. As described above, the implantation of a stent induces an inflammatory response causing the recruitment of monocytes to the stent site [117,118,119]. In a similar manner, monocytes are captured onto the vessel surface, where they undergo rolling and transmigration due to dysfunctional endothelial cells and differentiate into macrophages. Within the neointima, the macrophages become foam cells upon uptake of oxidised LDL [120]. Foam cell build-up within the stented region leads to plaque formation, with the subsequent development of a necrotic core of apoptotic macrophages, which is triggered, in part, by oxidised LDL [121]. In turn, these apoptotic macrophages are not cleared efficiently, which further enhances the inflammatory response [122]. Indeed, neoatherosclerosis is now considered a chronic inflammatory response to the stent itself. The resulting plaque can become calcified with or without a thin fibrous cap. The cap can promote thrombus development that can break off, resulting in severe complications such as myocardial infarction.

Neoatherosclerosis has emerged as a relatively new problem of PCI for which new stent technologies and/or drugs remain to be trialled and implemented in the clinic. It is not yet known whether current approaches to reducing atherosclerosis may be applicable to neoatherosclerosis associated with stenting. Recent evidence from the CANTOS trial showed reductions in cardiovascular events and inflammation using the monoclonal antibody canakinumab [123], an IL-1β inhibitor, whist colchicine, another anti-inflammatory agent, has shown promise in the COLCOT (Colchicine Cardiovascular Outcomes Trial) trial [124]. Reducing the inflammation long-term via specific anti-inflammatory drugs such as colchicine or through the use of monoclonal anti-inflammatory therapy has not been extensively studied either in preclinical models or in PCI patients, but may offer hope of diminished neoatherosclerosis formation in PCI patients.

## 3. Thrombosis

Thrombosis associated with bare metal and DES continues to be a major risk factor for myocardial infarction after stent placement. Endothelial cells play a large role in the prevention of thrombi formation. When the endothelium is intact, NO and PGI_2_ act synergistically to inhibit platelet activation and thus limit thrombus formation [23,125]. However, when ECs are denuded, the subendothelial collagen-rich ECM is exposed, and this leads to the attraction and activation of platelets [22,23]. The activation and subsequent adhesion of platelets are mediated by numerous glycoproteins (GPVI, GPIb/V/IX) and their interactions with von Willebrand factor (vWF) and the collagen-rich ECM [126,127]. Platelets then form a monolayer, which attracts other circulating platelets to form a 3D aggregate. Platelet aggregation occurs when GPIIb/IIIa integrins interact with thromboxane A2, ADP and ultra-large vWF [22,128].

Platelet-endothelial cell adhesion molecule-1 (PECAM-1/CD31) is a member of the immunoglobulin (Ig) gene superfamily [82]. It is expressed on both leukocytes and endothelial cells, and the homophilic interaction of the two PECAM-1s is required for leukocyte transmigration through the endothelial barrier [129]. JAMs (as mentioned earlier) and another protein called endothelial cell-selective adhesion molecule (ESAM) form one part of the endothelial tight junctions, with claudins and occludin also being a part of the tight junction at the apical region of the EC [130]. Tight junctions regulate the trafficking of ions and other small biomolecules amongst the cellular space. In contrast, vascular endothelial cadherin (VE-cadherin) and nectin form the adherens junctions in the centre of the intercellular barrier and control the flow of larger molecules [129,130]. Blocking of VE-cadherin with a monoclonal antibody was shown to increase vascular permeability and allow for neutrophils to pass the intercellular barrier [131]. This finding suggests the importance of VE-cadherin in maintaining the integrity of the endothelial barrier. Additionally, VE-cadherin (amongst other adhesion molecules) also plays a role in contact inhibition of ECs and their subsequent quiescence [130].

Platelets adhere to the site of injury utilising different mechanisms. These are: (1) recognition of ECM collagen by integrin α2β1 and GPVI [78] or (2) attachment via P-selectin glycoprotein ligand-1 (PSGL-1) and GPIbα [83]. Once platelets adhere to the inflamed tissue, they secrete P-selectin from alpha granules [20,83]. Another mechanism of platelet aggregation occurs in the setting of high shear stress. ECs activated by high shear stress can secrete large multimers of vWF, which form strings of ultra-large vWF. These tendrils capture circulating platelets, leading to their activation and adhesion to the endothelial layer [128,132]. In this circumstance, endothelial injury is not required, and thus, at sites of atherosclerotic plaque where high shear stress occurs, this mechanism is favoured over more classical EC injury pathways [128].

Regarding late stent thrombosis (LST) and very late stent thrombosis (VLST), the underlying mechanisms appear to be related to stent placement, and an increase in ROS is shown to be associated with stent thrombosis [133] that occurs post-placement. Furthermore, inflammation can trigger thrombosis, and conversely, thrombosis amplifies inflammation [134]. This chronic inflammation has been found to be due to the biomaterial interactions with blood leading to protein adsorption, activation and adhesion of platelets and activation of the coagulation cascade as well as inflammation pathways interlinked to these thrombogenic factors, such as the complement system [135]. C-reactive protein (CRP) has also been shown to be involved as a result of this chronic inflammation which initiates the coagulation cascade; as such, elevated CRP levels can be used as an indicator of stent thrombus formation [136].

Studies have found that, as assessed by optical coherence tomography (OCT), stent malapposition (lack of contact between a stent strut and the intimal surface), neoatherosclerosis, uncovered struts and stent under-expansion were all higher in patients with LST and VLST [137]. Commonly, the underlying cause of LST and VLST was found to be due to multiple factors, with malapposition alongside uncovered struts being the most common, the precise mechanisms of which are not yet well understood but may be due to the tissue defect surrounding the uncovered struts [138]. However, plaque rupture and occlusive restenosis have been found to contribute [137,139]. These findings suggest that both mechanisms of thrombi formation outlined above can be caused by multiple factors following PCI and stent placement and can cause severe complications if not treated appropriately.

## 4. Stents and Treatments Currently in the Clinic

Currently, there is a range of stents available to patients in clinical settings, and they are designed to specifically treat a range of complications that arise after stent implantation. These are described below, including more specific stents and the design elements implemented to improve patient outcomes.

### 4.1. Treatment Strategies

Each of the complications described above vary in terms of both the time it takes to affect the patient and the rate at which these occur. These variables also change depending on the treatment. The restenosis rate, with and without a BMS placement, following balloon angioplasty was studied in the BENESTENT and STRESS trials. In the BENESTENT trial, there was a significant decrease in the restenosis rate of the stent group when compared with the angioplasty alone group (22% stent, 32% angioplasty, *p* = 0.02) [140]. This was supported by the STRESS trial, with a significant reduction in the restenosis rate of 31.6% versus 42.1% (stent vs. angioplasty, respectively, *p* = 0.046) [141]. In each of these studies, the incidence of restenosis post-intervention decreased by approximately 10% when comparing the two groups. However, stent failure due to restenosis was still high, affecting 20–30% of patients at follow-up [140,141]. These outcomes led to the development of DESs, which secrete anti-proliferative drugs such as sirolimus. When comparing the restenosis of BMS and DES, it was noted that the composition of the neointima was slightly different. In BMS restenosis, the neointima consisted of a proteoglycan matrix with a high proportion of VSMCs, whilst in DES restenosis, the neointima was comprised of a proteoglycan-rich matrix but with lower numbers of VSMCs [8]. BMS restenosis usually presented early, around 6 months post-PCI, whereas DES restenosis presented later, i.e., around the 1–2-year mark [6,8]. Whilst the use of DES to treat NIH reduced the rates of restenosis markedly from ~30% to as low as 5%, patients with DM have a 2–4-fold increased risk of developing restenosis due to their unique vascular milieu [9]. Importantly, the anti-proliferative drugs used in DES have reduced the burden of NIH and ISR, but they have prolonged the duration of re-endothelialisation due to their lack of specificity in cell targeting. Thus, there is still the need to develop new stent coatings to mitigate these re-endothelialisation issues and to improve clinical outcomes.

When comparing neoatherosclerotic changes using autopsy data, early changes were seen in more patients who received a DES over a BMS (35% vs. 10%, *p* = 0.0004), with foamy macrophages being present as early as 4 months post-DES implantation [142]. In contrast, atherosclerotic changes with BMS occurred around 2 years, with it being a rare occurrence at 4 years post-implantation [142]. Despite these early changes, late restenosis (>1 year) is usually associated with neoatherosclerosis, whilst early restenosis (<1 year) is typically attributed to neointimal hyperplasia [143]. According to a recent optical coherence tomography (OCT) study, approximately 27.6% of very late DES stent failures were associated with neoatherosclerosis when examining both first- and newer-generation DES [137]. Treating neoatherosclerosis is not straightforward, and the stents currently on the market do not actively target it. One of the main strategies to prevent the development of neoatherosclerosis is risk minimisation. These risks could be patient-related, periprocedural or disturbed vessel haemodynamics [144]. As mentioned earlier, newer approaches such as those used in the CANTOS [123] or COLCOT trials [124] to target chronic inflammation are unique strategies that could be applied to the issue of in-stent neoatherosclerosis and are worthy of future investigations.

As for thrombosis, first-generation anti-proliferative DESs developed to combat restenosis had an unexpected increase in both late stent thrombosis (LST) (>30 days to 1 year) and very late stent thrombosis (VLST) (>1 year) despite the use of anti-platelet therapy [145]. Initially, the increased incidence of LST/VLST was attributed to delayed re-endothelialisation and under-expansion of the stent [146]. More recently, other possible causes in the development of stent thrombosis have been suggested, such as stent malapposition and neoatherosclerosis as evidenced by three OCT studies: Taniwaki et al. (*n* = 57) [137], PESTO French Registry (*n* = 97) [147] and PRESTIGE Registry (*n* = 134) [148].

#### 4.1.1. Dual Antiplatelet Therapy (DAPT)

The main treatment for thrombosis is dual antiplatelet therapy (DAPT). This is the combination of two drugs: one being aspirin and the other a P2Y_12_ inhibitor such as clopidogrel. These drugs bind to the P2Y_12_ receptor on platelets. This prevents the binding of ADP released by the platelet granules, which prevents further platelet aggregation [149]. Generally, administration of the P2Y_12_ inhibitor is withdrawn after 6–12 months, while aspirin is taken continuously; however, this timeline is largely dependent on the individual bleeding risk of each patient [150]. Aspirin, or acetylsalicylic acid, inhibits the production of prostaglandin E2 (PGE2) by the inhibition of COX-1, thereby preventing the production of thromboxane A_2_, the precursor to PGE2. Prostaglandin E2 is one of the molecules responsible for thrombus formation and is found in circulating blood following the activation of platelets [21,151,152].

#### 4.1.2. Paclitaxel

One of the early drugs utilised in DES to prevent NIH was paclitaxel. This drug polymerises and stabilises the microtubules within the cell, causing dysfunction, leading to cellular arrest at the pre-mitotic G_2_ phase or the mitotic phase of the cell cycle [153,154]. While initially, this drug alleviated the risk of NIH, it has been found to be less effective in clinical outcomes when compared to the -limus based drugs such as sirolimus and everolimus. This includes target lesion failure (4.2% vs. 6.8%) myocardial infarction (1.9% vs. 3.1%) and thrombosis (0.17% vs. 0.85%) [155]. These outcomes may be in part due to the activation of the p53 apoptotic pathway leading to cell death [156] and the drug causing a delay in healing and persistent inflammation [157].

#### 4.1.3. -limus Based Drugs

The -limus group of drugs inhibit the mechanistic target of rapamycin (mTOR), which arrests the cell cycle during the G_1_ phase. The mTOR protein kinase will usually form two different complexes known as mTORC1 and mTORC2, each with their own diverse functions and substrate interactions [158]. The mTORC1 complex has been more widely studied and is mostly responsible for phosphorylating substrates such as S6 kinase 1 (S6K1), amongst others, whilst regulating translation and cell growth [158]. mTORC1 plays a vital role in the progression of the cell cycle from the G_1_ phase to the S phase [158,159]. Conversely, the function of mTORC2 is less clear, but it is thought to interact with insulin/insulin-like growth factor 1 (IGF-1) receptor signalling [158]. This occurs by the binding of the -limus drug to the intracellular receptor FKBP12, which acts to decrease p27^kip1^ degradation [160]. As a cyclin-dependant kinase inhibitor (CKI), this increase in p27^kip1^ will arrest the cell cycle in VSMCs [161]. This class of drug, including everolimus, zotarolimus and sirolimus, are the most common anti-proliferative drugs used in second-generation DES designs. The effectiveness of this drug class in reducing NIH has arisen due to the targeting of vascular cell proliferation, including that of the VSMCs. However, a clear limitation of these drugs is their lack of specificity, resulting in the cell cycle arrest of endothelial cells and a reduction in the ability of endothelial cells to repair denuded vessels [162]. This drug class has also been shown to impact oxidative stress pathways causing cellular senescence via downregulation of Sirt1, which triggers the oxidative stress-endothelial senescence response, thereby arresting endothelial cells in the G_1_ phase [163]. In summary, the anti-proliferative drugs used in DES have reduced the burden of NIH and ISR, but they prolong the duration of re-endothelialisation due to their lack of specificity in cell targeting.

### 4.2. Current Stent Designs

According to market research, the companies which hold the largest market share for coronary stents are Boston Scientific, Medtronic and Abbott [164]. Table 2 outlines the stents produced by these companies, including the design aspects of each stent. The stent types can be divided into three categories: first-generation bare-metal stents (BMS), current drug-eluting stents (DES) and bioresorbable vascular stents (BVS).

#### 4.2.1. Bare-Metal Stents

BMS were the first kind of stents used and consisted of a bare metal scaffold. Initial BMS consisted of three types, each with a unique property to target specific complications known to arise from the stenting procedure. The main differences between the three BMS were the types of metal used, the construction process and the design, which impacted the strut thickness, radiopacity, durability and strength. The REBEL stent was designed to achieve higher radial and axial strength with low stent recoil. This was achieved by being made of a Pt-Cr alloy instead of Co-Cr, enabling a design with wider peaks, additional connectors on the proximal end and helical 2 connectors in the stent body. These design elements combatted lesion revascularisation and stent recoil as well as increased radiopacity [167,168]. The Integrity BMS differed from other BMS in its continuous sinusoidal construction achieved by the wrapping and laser fusion of a single wire around a mandrel in a sinusoid configuration. This design allowed for better tracking and easier access to vessels, with a focus on distal and tortuous vessels and without compromising radial strength. Finally, another BMS was the Multi-Link Vision, which was found to be comparable to first-generation DES, except for its increased risk of target vessel revascularisation [169]. Interestingly, this stent remains ideal for patients who require a shorter duration of DAPT. However, DESs produce consistently better outcomes than BMS in terms of reduced susceptibility to myocardial infarction, target lesion revascularisation and all-cause mortality but not stent thrombosis [170].

#### 4.2.2. Drug-Eluting Stents (DES)

The DES from Boston Scientific, SYNERGY (Marlborough, MA, USA), utilises a Pt-Cr base coated with an abluminal bioabsorbable polymer (poly(lactic-co-glycolic acid); PLGA) which dissolves after four months, leaving behind the bare metal stent. This occurs after a three-month release of the drug everolimus to combat in-stent restenosis. The rationale for bioabsorbable PLGA is to limit the inflammation caused by a permanent polymer being present; in turn, the lack of inflammation then facilitates better re-endothelisation [171]. Furthermore, the rate of stent thrombosis and late stent thrombosis at the one-year mark was found to be non-inferior and even surpassed competitors on the market [172,173]. This stent has also been used to treat high bleed-risk patients, given its shorter DAPT requirement due to the fast-absorbing polymer [174].

Medtronic Resolute Onyx and Resolute Integrity stents share the same drug elution properties, yet the Resolute Onyx stent is designed with a Pt-Ir core intended to increase the stent’s radiopacity to assist with more accurate placement within the blood vessel. Both stents elute zotarolimus, an mTOR inhibitor, which is released from the hydrophobic region of the Biolinx polymer. These stents show significant neointimal coverage after 30 days [175]. This beneficial in-stent re-endothelisation reduces the risk of thrombosis. This translates into a reduced requirement for DAPT treatment, which is favourable for patients with high bleed risk [143]. This stent has also been shown to be superior to polymer-free DES [176,177], which may be due to the choice of polymer coating playing a vital role in the healing process.

Finally, Abbott’s XIENCE stent is a Co-Cr base with durable fluoropolymer coating that releases everolimus over the course of 120 days. This stent design facilitates re-endothelisation and has increased thrombo-resistance and improved healing properties, which decreases the need for extended DAPT, resulting in better outcomes for high bleed-risk patients. The durable polymer has been shown to have improved outcomes over biodegradable polymers with respect to acute stent thrombosis; however, this property disappears after one year [178]. Following a ten-year study, it was found that biodegradable polymer coatings eluting sirolimus and durable polymers eluting everolimus have comparable outcomes, while permanent polymer coatings eluting sirolimus had significantly increased rates of adverse cardiac events and stent thrombosis [179].

#### 4.2.3. Bioresorbable Vascular Scaffolds (BVS)

BVSs are one of the newest innovations on the stent market, designed to elicit comparable short-term outcomes when compared to DESs but with more effective long-term outcomes, targeting LST and VLST. This is achieved by the design of the stent, which is fully resorbable, thus returning the artery to a more natural state, thereby reducing chronic inflammation associated with the permanent presence of a biomaterial in the artery. Initial results were promising in the ABSORB study with respect to restenosis, vasomotion and ST after two years with the scaffold being fully resorbed [180]. However, follow up meta-analysis of randomised controlled trials showed that there was a higher rate of subacute ST (24 h–1 month following implantation) in patients with a BVS when compared to those with second-generation DESs (3.11 [1.24–7.82]; *p* = 0.02) [181]. This is thought to be due to differences in design since bioresorbable materials need to have thicker struts to achieve the same radial force. This results in less flexibility, lower tensile strength, reduced manoeuvrability and fewer diameter sizes [182]. Furthermore, the bioabsorption of the scaffold and its impact on inflammation and any complications are still to be fully delineated. While these issues make current BVSs less desirable than comparable DESs, the areas outlined above can be considered design targets for next-generation BVSs which could see them replace DESs as the preferred device. Taken together, current stents and their clinical outcomes suggest that further improvements are required with a focus on enhancing re-endothelisation, which is key to better clinical outcomes.

## 5. Current and Future Directions of Stent Research

Current generation DESs on the market have been demonstrated to mostly reduce the clinical burden of ISR, ST and, therefore, the need for repeated re-vascularisation after restenosis. However, there is still a clinical need to develop novel stents that are able to actively target the causative mechanisms of restenosis and thrombosis in patients with complex vasculopathies. Whilst new materials with improved biocompatibility and radial strength have been a focus of stent research, recent work has focused on coatings with altered surface chemistry to provide enhanced healing, better cell attachment or surface immobilisation of bioactive molecules to elicit a beneficial anti-thrombotic or anti-restenotic response [183]. These coatings are ideally designed to facilitate the return of the artery to a more natural state (Figure 3 shows one example), and this would include eventual complete strut coverage. This has been defined as a layer of ECs attached to two layers of VSMCs to regain vascular integrity, thereby promoting barrier function and providing an anchor point for the ECs [184].

There are currently many different surface modification and material strategies under investigation, including thin film coatings, surface patterning, nanotexturing, liposomes, nanoparticles, nanotubes and bioconjugation of antibodies for cell capture [65,185,186,187,188]. An avenue to limit thrombosis and NIH is endothelial progenitor cell (EPC) capture using antibodies such as anti-CD34 and anti-CD146. While the initial use of the CD34 antibody was ineffective [189], more recently, the combined approach of CD146 antibodies to target late EPCs and silicone nanofibers to better capture the circulating cells is proving more efficacious [190]. Another approach is the direct placement of endothelial cells on the stent prior to implantation. This has been trialled in rabbits using altered endothelial cells to upregulate vascular endothelial growth factor (VEGF) to encourage re-endothelisation. While the overexpression of VEGF did reduce the incidence of restenosis, most transplanted cells did not remain on the stent during implantation. This may have arisen as a consequence of stent deployment or due to the animal’s immune response to the transplanted cells eliminating them from the stent. However, the VEGF produced in the initial stages while some of the cells were still attached was sufficient to begin the re-endothelisation process by the rabbits’ own ECs [191]. Taking advantage of the body’s natural healing processes alongside utilising some of the therapeutic molecules discussed throughout this review, such as GPx1 mimetics and NO donors, many of which have shown promising initial data, may be of use in alleviating ISR and ST, although further examination is required to determine if any of these strategies, in particular, could be translated to the clinic.

## 6. Conclusions

In this review, the causes and molecular mechanisms associated with restenosis and thrombosis have been discussed. In addition, we have highlighted the currently available and most widely used stents on the market, which do indeed mitigate many of the complications following PCI, except for those associated with pre-existing conditions that contribute to endothelial dysfunction such as diabetes mellitus. NIH is treated with drug-eluting stents that prevent VSMC proliferation, whilst thrombosis is treated with the use of DAPT. However, all currently available stents have limitations with respect to the treatment of diabetic patients with enhanced endothelial dysfunction, leading to a higher susceptibility to NIH and thrombosis. With predictions of growing numbers of diabetic patients over the next 20 years, newer mechanism-based approaches are urgently needed to target the known drivers of NIH, neoatherosclerosis and thrombosis. Targeting redox stress and inflammation following stenting is one strategy worth pursuing. Replenishing endothelial functionality, whether that be via improving NO bioavailability or through the replacement of lost endothelial cells, would restrict neoatherosclerosis formation and thrombosis. Importantly, the reduced reliance upon cell cycle inhibitory drugs currently used in DESs would allow for the re-endothelisation of stent surfaces and a return to a more natural state following the healing process. This, in turn, would reduce the likelihood of the occurrence of the three most debilitating complications (NIH, thrombosis and neoatherosclerosis) and increase the likelihood of positive outcomes for all patients, including those with diabetes mellitus.

## Figures and Tables

**Figure 1 biomolecules-12-00430-f001:**
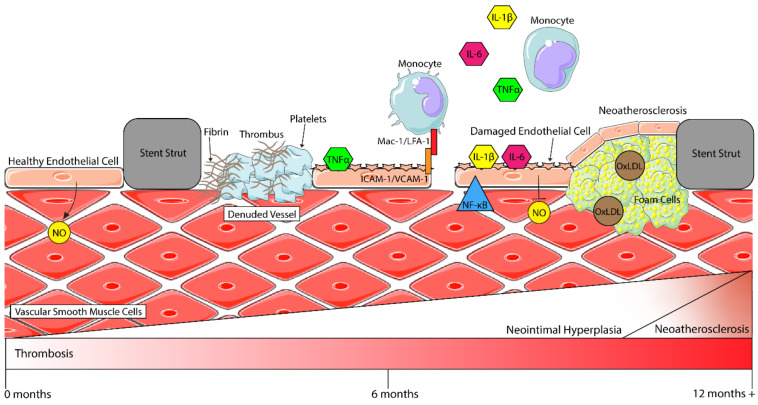
Inflammatory processes in the vasculature after stenting. Healthy endothelial cells (ECs) secrete nitric oxide (NO), which then interacts with vascular smooth muscle cells (VSMCs) to modulate vasoconstriction. After stenting, ECs are denuded from the vessel wall, and pro-inflammatory cytokines such as IL-6, TNFα and IL-1β are secreted by circulating monocytes and activate remaining ECs. More monocytes are then recruited to the site of injury by these cytokines and, through the interaction of Mac-1/LFA-1 with ICAM-1, begin firm rolling and attachment to the endothelium before transmigration into the subintimal space. When activated, ECs attract platelets to the stented region, where they begin to form a thrombus (clot) with fibrin molecules. Activated ECs also begin expressing NF-κB. This, in turn, downregulates NO production, and in combination with other factors, activates VSMCs, which begin to shift to a highly proliferative and migratory phenotype, thereby contributing to neointimal hyperplasia and restenosis. A late process (around 1–2 years) is neoatherosclerosis, where macrophages take up oxidised-LDL (Ox-LDL) to form foam cells which contribute to the development of an atherosclerotic plaque (figure assets from Servier Medical Art—smart.servier.com).

**Figure 3 biomolecules-12-00430-f003:**
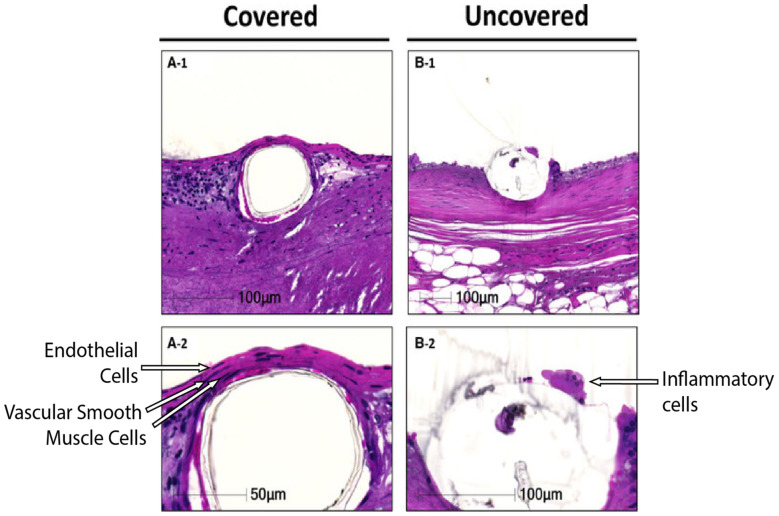
Covered and uncovered stent struts. (**A-1**), strut coverage with 2 layers of vascular smooth muscle cells and a monolayer of endothelial cells. (**B-1**), uncovered strut with coverage of inflammatory cells. (**A-2**,**B-2**) higher-powered images highlighting strut coverage (Adapted from [184]).

**Table 2 biomolecules-12-00430-t002:** Most commonly utilised stents according to market share.

Stent (Manufacturer)	Material	Coating	Elution Mechanism	Drug	Release Time Frame	Reference
**Boston Scientific**	
SYNERGY	Pt-Cr	Abluminal Bioabsorbable Polymer (PLGA)		Everolimus	3 month drug release, 4 month polymer absorption	Wilson et al. [165]
REBEL	Pt-Cr	BMS	N/A	N/A	N/A	
**Medtronic**	
Resolute Onyx	Co-Cr, PtIr Core	BioLinx™ polymer	Drug released from hydrophobic section of BioLinx polymer	Zotarolimus		Jinnouchi et al. [143]
Resolute Integrity	Co-Cr	BioLinx™ polymer	Drug released from hydrophobic section of BioLinx polymer	Zotarolimus		Jinnouchi et al. [143]
Integrity	Co-Cr	BMS	N/A	N/A	N/A	
**Abbott**	
XIENCE	Co-Cr	Fluoropolymer™		Everolimus	120 days	Kukreja et al. [166]
Multi-Link Vision	Co-Cr	BMS	N/A	N/A	N/A

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
