# Peer review of "The Mechanisms of Restenosis and Relevance to Next Generation Stent Design"

_biomolecules, 2022, doi:10.3390/biom12030430_

Round 1

Reviewer 1 Report

Stents are life saving devices. Unfortunately, until now, none of the scientific literature has demonstrated the evidence of a stent with no late complications such as restenosis or thrombosis incidence, vascular inflammation, among others.

These complications are a multifactorial problem related to patient and comorbidities, lesion characteristics and location, atherosclerosis burden, procedural factors and to the individual response to therapy.

To develop more efficient platforms in terms of clinical efficacy whether involving controlled drug or mediators release and capture of relevant cells it is important to unravel the major actors and mechanisms that govern these processes in the vasculature after stenting. Only with a profound knowledge of the mechanisms involved in the disease and late complications after stenting significant advances in the field are expected.

In fact, this aspect is the element that brings novelty to this review. It goes beyond the usual field of clinical outcomes of restenosis after stenting. It covers very important issues of vascular inflammation, endothelial dysfunction and neoatherosclerosis.

Authors overview a variety of studies focusing the role of inflammatory cells and vascular cells in the intricate pathways that may lead to late stent complications. A particular emphasis was given to inflammatory molecules, signalling pathways and mediators of the redox balance that may be relevant in patients with diabetes undergoing PCI and stent implantation.

The focus on diabetes is important. Nowadays, diabetes is a comorbidity of major concern for CAD.  Regrettably obesity was not focused in the review as the prevalence of diabetes and CAD is growing mainly due to the incidence of obesity. In this context, the relationship of adipose tissue with vascular inflammation and endothelial dysfunction, should be considered.

The review is very comprehensive. It is supported by recent, extensive and relevant bibliography.

From my point of view this is a very interesting paper that will be very useful for professionals in different scientific areas from engineering and biotechnology to medicine.   

Author Response

Response to Reviewers

Reviewer 1:

Comment: Stents are life saving devices. Unfortunately, until now, none of the scientific literature has demonstrated the evidence of a stent with no late complications such as restenosis or thrombosis incidence, vascular inflammation, among others.

These complications are a multifactorial problem related to patient and comorbidities, lesion characteristics and location, atherosclerosis burden, procedural factors and to the individual response to therapy.

To develop more efficient platforms in terms of clinical efficacy whether involving controlled drug or mediators release and capture of relevant cells it is important to unravel the major actors and mechanisms that govern these processes in the vasculature after stenting. Only with a profound knowledge of the mechanisms involved in the disease and late complications after stenting significant advances in the field are expected.

In fact, this aspect is the element that brings novelty to this review. It goes beyond the usual field of clinical outcomes of restenosis after stenting. It covers very important issues of vascular inflammation, endothelial dysfunction and neoatherosclerosis.

Authors overview a variety of studies focusing the role of inflammatory cells and vascular cells in the intricate pathways that may lead to late stent complications. A particular emphasis was given to inflammatory molecules, signalling pathways and mediators of the redox balance that may be relevant in patients with diabetes undergoing PCI and stent implantation.
Author Response: We thank the reviewer for their kind comments on our paper.

Comment: The focus on diabetes is important. Nowadays, diabetes is a comorbidity of major concern for CAD.  Regrettably obesity was not focused in the review as the prevalence of diabetes and CAD is growing mainly due to the incidence of obesity. In this context, the relationship of adipose tissue with vascular inflammation and endothelial dysfunction, should be considered.
Author Response: Agree, we have added the following text about perivascular adipose tissue and obesity to the manuscript from lines 213-229 on page 5.

Comment: The review is very comprehensive. It is supported by recent, extensive and relevant bibliography.

From my point of view this is a very interesting paper that will be very useful for professionals in different scientific areas from engineering and biotechnology to medicine.
Author Response: We thank reviewer 1 for their comments and feedback on the paper.    

Reviewer 2 Report

This review by Clare et al is very comprehensive and covers most important topics on in stent restenosis.

The manuscript reads very well and describes the most common types of stents and the various phases of revascularization after stent placement in detail.

Some minor comments:

In the paragraph on vascular inflammation I miss the role of CCL2 and especially the contribution of toll like receptors as activators of the NF-κB signaling pathway.

In paragraph 3.2 Refer to  specific EC specific integrins /cadherins  and their role in leukocyte transmigration which are disrupted upon endothelial denudation.

Line 182/183 1, ICAM-1 and E-selectin) [33] and a chemokine (monocyte chemoattractant protein-1; 182 MCP-1/CCL2) were upregulated. MCP-1 is already mentioned for the first time in line 174.

The role of TGF beta is more complex than now described in line 349-350 and should be addressed in more detail.

Line 371-378 This paragraph is redundant in the light of the paragraphs on endothelial dysfunction described under 2.3.

In paragraph 2.4 the contribution of Endo-MT to neointimal hyperplasia should be briefly described.

Author Response

Reviewer 2:

Comment: This review by Clare et al is very comprehensive and covers most important topics on in stent restenosis.

The manuscript reads very well and describes the most common types of stents and the various phases of revascularization after stent placement in detail.

Author Response: We thank reviewer 2 for their feedback about our manuscript.

Comment: In the paragraph on vascular inflammation I miss the role of CCL2 and especially the contribution of toll like receptors as activators of the NF-κB signaling pathway.

Author Response: We included the below text from lines 166 to 175 on page 4 of the revised manuscript in response to the contribution of toll-like receptors as activators of NF-κB.

We also included the following text to clarify the role of MCP-1/CCL2, on lines 205-209 of page 5.

Comment: In paragraph 3.2 Refer to  specific EC specific integrins /cadherins  and their role in leukocyte transmigration which are disrupted upon endothelial denudation.

Author Response: We added the following text from lines 476 to 490 on page 12 to clarify some specific EC adhesion molecules.

Comment: Line 182/183 1, ICAM-1 and E-selectin) [33] and a chemokine (monocyte chemoattractant protein-1; 182 MCP-1/CCL2) were upregulated. MCP-1 is already mentioned for the first time in line 174.
Author Response: Thank you, we removed the following text from line 193 “monocyte chemoattractant protein-1;” and added it to line 183 after the first mention of MCP-1.

Comment: The role of TGF beta is more complex than now described in line 349-350 and should be addressed in more detail.
Author Response: We added the following from lines 381-395 to elaborate on the role of TGF beta.

Comment: Line 371-378 This paragraph is redundant in the light of the paragraphs on endothelial dysfunction described under 2.3.
Author Response:
We removed “Endothelial damage is caused by the stenting procedure itself, exposing VSMCs to circulating blood elements, including platelet and leukocyte responses which cascade, as described above, into NIH if left untreated. NO is expressed by endothelial cells and acts as a vasodilator, anti-proliferative [81] and anti-adhesive agent [82], [83]. Following vascular injury, endothelial dysfunction results in impairment of NO production, which is exacerbated by conditions such as DM [84].”

We kept the remaining sentence from lines 414-416 on page 9 (“While endothelial dysfunction and damage is the primary driver behind NIH, adventitial fibroblasts are also recognised to play a significant role in NIH”) of the revised manuscript and moved them to the end of the previous paragraph.

Comment: In paragraph 2.4 the contribution of Endo-MT to neointimal hyperplasia should be briefly described.
Author Response: To address the contribution of Endo-MT to NIH the following text was added to lines 390-394.

Additional Comments: Apart from those mentioned above, we also made some minor revisions to grammar, spelling and punctuation as indicated by the tracked changes in the revised manuscript.